FLASC: a flare-sensitive clustering algorithm

Bot Daniël M. 1
Peeters Jannes 1
Liesenborgs Jori 2
http://orcid.org/0000-0002-6416-2717 Aerts Jan 3 jan.aerts@kuleuven.be
1 Data Science Institute (DSI), Universiteit Hasselt , Diepenbeek , Belgium
2 Expertisecentrum voor Digitale Media (EDM), Flanders Make, Universiteit Hasselt , Diepenbeek , Belgium
3 Augmented Intelligence for Data Analytics (AIDA) Lab, Department of Biosystems, Katholieke Universiteit Leuven , Leuven , Belgium
Datta Anwitaman
Electronic publication date: 2025 Apr 18
Publication date: 2025
Volume: 11
Electronic Location ID: e2792
Received 2024 Sep 4; Accepted 2025 Mar 10
Copyright: © 2025 Bot et al.
Copyright year: 2025
Copyright holder: Bot et al.
License: This is an open access article distributed under the terms of the Creative Commons Attribution License, which permits unrestricted use, distribution, reproduction and adaptation in any medium and for any purpose provided that it is properly attributed. For attribution, the original author(s), title, publication source (PeerJ Computer Science) and either DOI or URL of the article must be cited.
License URL: https://creativecommons.org/licenses/by/4.0/

Keywords: Exploratory data analysis, Density-based clustering, Branch-hierarchy detection, HDBSCAN*

Funding: KU Leuven STG/23/040 Hasselt University BOF BOF20OWB33, BOF21DOC19 This work was supported by KU Leuven grant STG/23/040 and Hasselt University BOF grants (BOF20OWB33) and (BOF21DOC19). The funders had no role in study design, data collection and analysis, decision to publish, or preparation of the manuscript.

==============================
Exploratory data analysis workflows often use clustering algorithms to find groups of similar data points. The shape of these clusters can provide meaningful information about the data. For example, a Y-shaped cluster might represent an evolving process with two distinct outcomes. This article presents flare-sensitive clustering (FLASC), an algorithm that detects branches within clusters to identify such shape-based subgroups. FLASC builds upon HDBSCAN*—a state-of-the-art density-based clustering algorithm—and detects branches in a post-processing step using within-cluster connectivity. Two algorithm variants are presented, which trade computational cost for noise robustness. We show that both variants scale similarly to HDBSCAN* regarding computational cost and provide similar outputs across repeated runs. In addition, we demonstrate the benefit of branch detection on two real-world data sets. Our implementation is included in the hdbscan Python package and available as a standalone package at https://github.com/vda-lab/pyflasc.

Introduction

Exploratory data analysis (EDA)—i.e., searching for interesting patterns in data—is ubiquitous in data science and knowledge discovery workflows. Detecting which groups of similar observations exist is a common step in EDA. Typically, such groups are detected as clusters. Several early clustering algorithms—such as kMeans and average-linkage hierarchical clustering—rely on a minimum variance principle, restricting them to finding clusters with convex shapes (Cormack, 1971; Campello et al., 2015). Density-based clustering algorithms do not have this limitation (Campello et al., 2015). Informally, they specify clusters as regions of high density separated by regions of lower density, allowing them to capture cluster shapes. The shape of a cluster can reveal additional relevant subgroups. For example, a Y-shaped cluster might represent an evolving process with two distinct outcomes. Consequently, the branches in a cluster’s manifold—i.e., flares—can represent meaningful subpopulations in datasets (see, e.g., Reaven & Miller, 1979; Lum et al., 2013; Kamruzzaman, Kalyanaraman & Krishnamoorthy, 2018; Skaf & Laubenbacher, 2022).

Clustering algorithms generally cannot detect this type of subgroup because no gap separates flares from their cluster. From a topological perspective, clustering algorithms describe the connected components in a simplicial complex of the data (Carlsson, 2014): a set of points, edges, and triangles that describe connectivity. Flares—i.e., branches in a cluster’s manifold—are connected in the simplicial complex. In other words, there is a path between data points in different branches that exclusively goes through data points ‘that lie close together’. Therefore, they have a vanishing homology and cannot be detected as persistent clusters.

Several flare-detection techniques have been proposed in topological data analysis literature. For example, Carlsson (2014) proposed functional persistence to distinguish flares from a data set’s central core. This technique quantifies data point centrality as the sum of its distances. Central observations have lower distance sums than points towards the extreme ends of the feature space. A manually controlled centrality threshold removes the data’s core, separating branches from each other and making them detectable as (density-based) clusters.

While Carlsson’s functional persistence can detect branches as clusters, the single centrality threshold cannot describe how branches merge into each other as more central points are included. The cluster hierarchy that a moving centrality threshold would form is analogous to a density-contour tree (Hartigan, 1975) as used in Hierarchical DBSCAN* (HDBSCAN*) (Campello et al., 2015) to replace Density Based Spatial Clustering of Applications with Noise (DBSCAN)’s (Ester et al., 1996) density threshold. Instead of describing which clusters exist over the density range, it models connectivity over centrality, and its centrality-contour clusters correspond to branches rather than clusters. We will refer to the centrality-contour tree as a branching hierarchy.

Extending functional persistence to construct branching hierarchies requires a process that considers all centrality and distance thresholds, called bi-filtration. The centrality controls how much of the core is retained to describe how branches grow and merge, and the data point distances determine whether points are connected and form a cluster. Algorithms for extracting persistent structures from bi-filtrations are computationally expensive (Lesnick & Wright, 2022; Kerber & Rolle, 2021). Their resulting bi-graded hierarchies are also complicated to work with, as they do not have a compact representation (Carlsson, 2014) (research into usable representations is ongoing (Botnan et al., 2022)), and existing visualisations are non-trivial (Lesnick & Wright, 2015; Scoccola & Rolle, 2023). Alternative strategies simultaneously vary both dimensions in a single-parameter filtration (Chazal et al., 2009); however, they remain computationally expensive (Vandaele et al., 2021).

The present article presents an approach that efficiently computes branching hierarchies and detects branch-based subgroups of clusters in unfamiliar data. Inspired by Vandaele et al. (2021), we compute the branching hierarchies using graph approximations of the data. This technique effectively replaces functional persistence’s manual centrality threshold with a question: which data points should be connected in the approximation graph? We will use HDBSCAN* (Campello, Moulavi & Sander, 2013; Campello et al., 2015)—a state-of-the-art density-based clustering algorithm—to answer this question. Conceptually, our approach can be thought of as creating a sequence of subgraphs that progressively include more and more central points and tracking the remaining connected components. Interestingly, a similar method has been used by Li et al. (2017) to detect actual branches in 3D models of plants.

Our main contribution is this flare detection approach, implemented as a post-processing step in the HDBSCAN* implementations by McInnes, Healy & Astels (2017; hdbscan: https://github.com/scikit-learn-contrib/hdbscan; fast_hdbscan: https://github.com/TutteInstitute/fast_hdbscan) and as a stand-alone package (pyflasc: https://github.com/vda-lab/pyflasc). We propose two types of approximation graphs that naturally arise from HDBSCAN*’s design and provide a practical centrality measure that is computable in linear complexity. Combining density-based clustering and flare detection into a single algorithm provides several attractive properties: The ability to detect clusters and their branches.

Intuitive minimum cluster and branch size parameters rather than density and centrality thresholds (McInnes & Healy, 2017).

Low computational cost compared to multi-parameter persistence and other structure learning algorithms.

High branch-detection sensitivity and noise robustness by operating on HDBSCAN*-clusters, thereby suppressing spurious noisy connectivity.

Branch detection at multiple distance scales because each cluster has a separate approximation graph.

We call the resulting algorithm flare-sensitive clustering (FLASC) and empirically analyse its computational cost and stability on synthetic data sets to show that the flare detection cost is relatively low. In addition, we demonstrate FLASC on two real-world data sets, illustrating its benefits for data exploration.

The remainder of this article is organised as follows: The ‘Related Work’ section provides a literature overview of related data analysis algorithms and describes the HDBSCAN* algorithm in more detail. The ‘The FLASC Algorithm’ section describes how FLASC builds on HDBSCAN* to detect branches within clusters and discusses the algorithm’s complexity and stability. The ‘Experiments’ section presents our empirical analyses demonstrating the algorithm’s computational complexity, stability, and benefits for data exploration. Finally, the ‘Discussion’ and ‘Conclusion’ sections discuss our results and present our conclusions. Portions of this text were previously published as part of a preprint: Bot et al. (2024).

Related work

The purpose of our work is to detect branching structures within clusters. As such, our work relates to manifold and structure learning algorithms in general. This section provides an overview of clustering and related data analysis algorithms. In addition, we introduce HDBSCAN*, the density-based clustering algorithm we build upon.

Clustering algorithms

Clustering algorithms have been reviewed on many occasions. For example, Ezugwu et al. (2022) list twelve other surveys to start their comprehensive review. A thought-provoking 1971 publication critically reviews clustering theory and practice, much of which remains relevant today (Cormack, 1971). For example, Cormack (1971) explains that clustering techniques are often based on conflicting ideas of what clusters are due to the absence of an agreed-upon formal definition. Generally, clusters are described as groups of similar observations dissimilar to others (Cormack, 1971; Xu & Wunsch, 2005; Ezugwu et al., 2022). Whether clusters should be spherical or if “multidimensional amoebae” shapes are acceptable was similarly contested at the time (Cormack, 1971).

Cormack’s categorisation of clustering algorithms is also still relevant. Generally, clustering algorithms are divided into those that rely on a cluster hierarchy and those that operate by optimising partitions (Xu & Wunsch, 2005; Ezugwu et al., 2022). Below, we provide a brief overview of these categories. We refer to Xu & Wunsch (2005) and Ezugwu et al. (2022) for more elaborate reviews of clustering research.

Hierarchical clustering

Hierarchical clustering algorithms are divided into agglomerative and divisive algorithms (Cormack, 1971; Xu & Wunsch, 2005; Ezugwu et al., 2022). Agglomerative algorithms build a dendrogram by successively merging the nearest points and the groups they create. The similarity between groups is defined by a linkage criterion such as single, average, complete, or ward linkage (Ward, 1963). We refer to Cormack (1971) for a detailed explanation of these criteria and references to earlier works. Interestingly, single linkage hierarchies are closely related to minimum spanning trees, allowing for more efficient computation (Cormack, 1971) and interpretation in topological data analysis terms (Carlsson, 2014). Divisive algorithms work in the opposite direction and look for dissimilarity values at which groups separate.

Density-based clustering algorithms are related to hierarchical clustering because the principles they rely on are inherently hierarchical (Hartigan, 1975). Density-based clusters are already described in Cormack (1971) as high-density regions in data space separated by low-density regions. Recent algorithms generally use density-contour clusters and density-contour trees formalised by Hartigan (1975) (as explained by Campello et al., 2015). For example, DBSCAN (Ester et al., 1996) defines clusters as connected components, where points are connected if they are within a specified distance ε of each other and have a minimum number of neighbours within ε. OPTICS (Ankerst et al., 1999) builds on DBSCAN by creating a reachability plot for visualising density profiles. HDBSCAN* (Campello et al., 2015; Campello, Moulavi & Sander, 2013) succeeds OPTICS by adapting DBSCAN with Hartigan’s principles and evaluating the full density-contour tree. Recent adaptations of these algorithms include block-guided DBSCAN (Xing & Zhao, 2024) and an approach to extract hybrid DBSCAN–HDBSCAN* clusters from the HDBSCAN* cluster hierarchy (Malzer & Baum, 2020).

Partitional clustering

Partitional clustering algorithms interpret clustering as an optimisation problem. For example, kMeans selects k initial cluster centres, which are then optimised to minimise the distances between points and their assigned cluster (MacQueen, 1967). A popular implementation (Pedregosa et al., 2011) supports the kMeans++ initialisation strategy (Arthur & Vassilvitskii, 2006) and an efficient optimisation algorithm by Lloyd (1982). Many variants of kMeans exist, such as kMediods (Park & Jun, 2009) and fuzzy c-Means (Bezdek, Ehrlich & Full, 1984).

Genetic algorithms have also been proposed for clustering (Xu & Wunsch, 2005; Ezugwu et al., 2022). They provide a search strategy inspired by evolution and natural selection for optimisation problems in general. Genetic algorithms share common drawbacks with search algorithms: they do not guarantee optimal solutions, are expensive to compute, and tend to require careful tuning of their parameters to result in useful solutions (Xu & Wunsch, 2005).

Other clustering algorithms

Many other clustering algorithms do not fit neatly into the hierarchical or partitional categories. Spectral clustering is such an algorithm. It uses the eigenvectors of a Laplacian matrix to project data into a space that reflects the data’s intrinsic structure (Belkin & Niyogi, 2003). Damle, Minden & Ying (2019) show how clusters can be directly extracted from this eigenvector space. Other dimensionality reduction algorithms—such as principal component analysis (PCA) (Pearson, 1901; Hotelling, 1936) and Uniform Manifold Approximation and Projection (UMAP) (McInnes, Healy & Melville, 2020)—are also used as preprocessing steps for clustering (as in, e.g., Packer et al., 2019).

The recently proposed clustering by measuring local direction centrality (CDC) algorithm by Peng et al. (2022) also does not fit into the previous categories. Instead, Peng et al. (2022) refer to it as a “boundary-seeking” clustering algorithm. CDC computes a direction centrality metric (DCM) measuring the spread of points’ k-nearest neighbours to find boundaries between clusters. Intuitively, the neighbours of points within a cluster are spread out, while the neighbours of points on the boundary mainly lie towards the cluster’s centre. This idea has previously been used to summarise the structure of weighted graphs (Vandaele, Saeys & De Bie, 2020). CDC considers the highest DCM points as boundary points. A manual ratio parameter controls how many points form the boundary. Distances to the nearest boundary point limit the connectivity used to extract clusters. CDC’s advantage over density-based clustering is that it can separate clusters connected through a few short edges (Peng et al., 2022).

Other algorithms are designed for tasks that require specific properties. For example, Sun et al. (2024) propose TWStream, a three-way clustering algorithm for data streams. Three-way clustering considers an addition boundary region indicating points with uncertain cluster membership. Fuzzy clustering algorithms also provide uncertain cluster memberships, allowing for observations that belong to multiple clusters (Xu & Wunsch, 2005; Ezugwu et al., 2022).

Böhm et al. (2004) propose another specialised clustering algorithm called 4C. It extracts points with linear correlations from density-based clusters. Consequently, 4C can detect flares that are linearly correlated in data space. Tung, Xu & Ooi (2005) expand on 4C with Curler, an algorithm that builds clusters with non-linear correlations from Gaussian micro-clusters computed with expectation maximisation (EM) (Dempster, Laird & Rubin, 1977). While Curler can detect non-linear relationships, it is not designed to detect branches in the micro-cluster graph it creates.

To our knowledge, no other clustering algorithm combines the benefits of FLASC with the ability to detect branching structures in clusters. In particular, FLASC does not require the number of clusters to be specified in advance, thereby avoiding a fundamental problem in evaluating cluster validity (Xu & Wunsch, 2005). In addition, FLASC is robust to noise and efficient to compute due to its reliance on HDBSCAN*. Furthermore, FLASC provides stable results across repeated runs and different samples of the same underlying distribution. Finally, FLASC can detect branches with points that are not linearly correlated.

Structure learning

Many data types lie not just on a manifold but on a smooth, one-dimensional manifold. Extracting such manifolds can be essential in unsupervised learning applications. For example, road networks can be extracted from GPS measurements (Bonnaire, Decelle & Aghanim, 2022), and cell developmental trajectories can be extracted from gene expression data (Qiu et al., 2017; Vandaele, Saeys & De Bie, 2020). Algorithms for extracting such structures are related to our work because the branch-based subgroups we are interested in can be extracted from them by partitioning the data between their intersections (Chervov, Bac & Zinovyev, 2020).

Most work on extracting smooth, one-dimensional manifolds is based on principal curves: a smooth, self-consistent curve that passes through the middle of the data (Hastie & Stuetzle, 1989). Techniques estimating principal curves, trees, or graphs are often based on expectation maximisation (Dempster, Laird & Rubin, 1977) and optimise the one-dimensional manifold directly (e.g., Bonnaire, Decelle & Aghanim, 2022; Mao et al., 2017). Alternative approaches are more closely related to non-linear dimensionality reduction (DR) algorithms that model the data’s structure as an undirected graph (e.g., Roweis & Saul, 2000; Tenenbaum, de Silva & Langford, 2000; Belkin & Niyogi, 2003; van der Maaten & Hinton, 2008; McInnes, Healy & Melville, 2020). For example, Vandaele, Saeys & De Bie (2020) use (manually) pruned minimum spanning trees over edges weighted by their boundary coefficient to extract a graph’s backbone. Alternatively, Ge et al. (2011) extract graph skeletons using a Reeb Graph. Reeb Graphs track connected components’ existence, merges, and splits in level sets of a continuous function defined on a manifold. Using geodesic distances to an arbitrary eccentric point as the continuous function makes the Reeb Graph capture the manifold’s skeleton. Interestingly, Mapper—an algorithm approximating Reeb Graphs (Singh, Memoli & Carlsson, 2007)—has also been used for detecting branch-based subpopulations (Kamruzzaman, Kalyanaraman & Krishnamoorthy, 2018).

There are several similarities between these methods and our work. Like Vandaele, Saeys & De Bie (2020), our approach detects tree-based branching hierarchies. Like Ge et al. (2011), our approach is topologically inspired. Where they create a Reeb Graph, we compute a join tree. The main difference between these methods and our work is their goal. We aim to identify relevant branch-based subpopulations. Ge et al. (2011), Vandaele, Saeys & De Bie (2020), and the expectation maximisation-based algorithms explicitly model the data’s structure, necessitating a higher computational cost.

HDBSCAN*

HDBSCAN* is a state-of-the-art density-based clustering algorithm (Campello, Moulavi & Sander, 2013; Campello et al., 2015). Informally, density-based clustering specifies clusters as regions of high density separated by regions of lower density. This formulation does not limit clusters to convex shapes and provides a natural way to separate noise points from clusters. The algorithm is well suited for exploring unfamiliar data because HDBSCAN* does not require the number of clusters or the distance between clusters to be specified in advance.

Several studies have implemented and adapted the HDBSCAN* algorithm. McInnes & Healy (2017) improved the algorithm’s computational performance by using space trees to find the data points’ nearest neighbours and provided an efficient Python implementation (McInnes, Healy & Astels, 2017). Stewart & Al-Khassaweneh (2022) created a Java implementation with a novel prediction technique for unseen data points. Jackson, Qiao & Xing (2018) presented an approximate HDBSCAN* algorithm that uses NN-descent (Dong, Moses & Li, 2011) to find the nearest neighbours with fast distributed performance. Malzer & Baum (2020) introduced a cluster selection distance threshold that effectively creates a hybrid between DBSCAN’s (Ester et al., 1996) and HDBSCAN*’s cluster selection, improving the algorithm’s performance on data sets with small clusters and a large density variability. Neto et al. (2021) showed how relative neighbourhood graphs (RNGs) (Toussaint, 1980) can be used to efficiently compute HDBSCAN* cluster hierarchies for multiple min. cluster size values. Their follow-up work presented MustaCHE, a visualisation tool for the resulting meta-cluster hierarchy (Neto et al., 2018). To our knowledge, no previous study has adapted HDBSCAN* for detecting flares.

Because our work builds on HDBSCAN*, it is relevant to explain how the algorithm works in more detail. The remainder of this section describes HDBSCAN* following Campello et al. (2015)’s explanation. We refer the reader to McInnes & Healy (2017) for a more formal, statistically motivated description.

The HDBSCAN* algorithm

HDBSCAN* is based on density-based clustering concepts pioneered by Wishart (1969) and formalised by Hartigan (1975). We demonstrate these ideas using a 2D point cloud adapted from McInnes, Healy & Astels (2022), shown in Fig. 1A. In general, let X={x1,…,xN} be a data set consisting of N feature vectors x(⋅) and a distance metric d(xi,xj). Then, HDBSCAN* estimates the density at point xi as (Campello et al., 2015):

(1) λk(xi)=1/κ(xi),

where the point’s core distance κ(xi) is the distance to its k-nearest neighbour.

Figure 1 Density-based clustering concepts behind HDBSCAN*.

(A) A 2D example point cloud with varying density adapted from McInnes, Healy & Astels’s (2022) online tutorial. (B) Density contours in a height map illustrate the data’s density profile. Peaks in this density profile correspond to density contour clusters. (C) Clusters extracted from the density profile by HDBSCAN* indicated in colour. (D) The density contour tree describes how density contour clusters merge when considering lower density thresholds.

Figure 1B illustrates the example’s density profile as contours in a height map. Density contour clusters intuitively correspond to peaks in the density profile, for example, the clusters indicated in colour in Fig. 1C. More formally, the density contour clusters at some threshold λt are a collection of maximal, connected subsets in a level set {x|λ(x)≥λt} (Hartigan, 1975). In other words, density contour clusters are the connected components of points with a density higher than some threshold. Density contour trees capture the hierarchy in which density contour clusters merge as the density threshold decreases. From a topological perspective, density contour trees are a join tree of the data’s density profile.

Data sets generally do not have an inherent notion of connectivity between their data points. Such connectivity is needed to determine whether two points are part of the same density contour cluster at a threshold λt. HDBSCAN* solves this problem by considering points to be connected if the distance between them is smaller than or equal to 1/λt. This solution is possible because density is defined in terms of distance in Eq. (1). HDBSCAN* uses a mutual reachability distance between points for this purpose, which is defined as (Campello et al., 2015):

(2) dmreach(xi,xj)={max{κ(xi),κ(xj),d(xi,xj)}ifxi≠xj,0otherwise,

where the value of k, as used in κ, is specified manually and acts as a smoothing factor for the density estimation.

We can now recover the density contour tree using the mutual reachability distance to provide connectivity. The edges that change the connectivity between density contour clusters are exactly those edges in the data’s Minimum Spanning Tree (MST) (as cited in Cormack, 1971). HDBSCAN* uses an MST to efficiently compute a single linkage clustering hierarchy (Sibson, 1973). The resulting dendrogram is simplified using a manually specified minimum cluster size mc to recover a condensed cluster hierarchy that resembles the data’s density profile, as shown in Fig. 1D. From the root down, only the sides of a split containing more than mc points are considered to represent clusters. Sides with fewer points are interpreted as “falling out of the parent cluster” (McInnes & Healy, 2017) or the cluster disappearing completely.

HDBSCAN* provides two strategies for selecting clusters from the condensed hierarchy: the excess of mass (EOM) strategy and the leaf strategy (Campello et al., 2015). The leaf strategy selects all leaf segments in the condensed hierarchy, typically resulting in multiple small clusters. The EOM strategy maximises relative cluster stability while preventing any data point from being a member of more than one selected cluster. A cluster Cj’s relative stability σk(Cj) is defined as Campello et al. (2015):

(3) σk(Cj)=∑xi∈Cjλk,maxCj(xi)−λk,minCj,

where λk,maxCj(xi) is the density at which xi falls out of Cj or Cj separates into two clusters, and λk,minCi is the minimum density at which Cj exists. In other words, the stability of a cluster is the sum of density ranges in which points are part of the cluster, corresponding to the area of the cluster’s icicle in Fig. 1D. HDBSCAN*’s cluster selection epsilon parameter can be used to specify a minimum persistence for EOM clusters (Malzer & Baum, 2020).

Flare-sensitive hdbscan*

Our work’s main contribution is a flare detection post-processing step for HDBSCAN*. This section describes how the post-processing step works and integrates with HDBSCAN* to form our FLASC algorithm (see Algorithm 1). FLASC starts by evaluating a flat HDBSCAN* clustering, keeping track of the space tree used in HDBSCAN* (McInnes, Healy & Astels, 2017; McInnes & Healy, 2017) to find nearest neighbours efficiently. One noteworthy change from McInnes, Healy & Astels (2017) is that we give all points the 0-label when a single cluster is allowed and selected, and the cluster selection epsilon parameter (Malzer & Baum, 2020) is not used. This change enables FLASC to better analyse branching structures in data sets that contain a single cluster. Then, a branch detection step is performed for each selected cluster Cj, explained in more detail below.

Algorithm 1 A high-level overview of the FLASC algorithm.

1:   function FLASC( X,d)	
   ▹ X is a dataset with N feature vectors x(⋅) and d is a distance metric d(xi,xj).	
2:      evaluate HDBSCAN (X,d) and store its internal data structures.	
3:     for each detected cluster Cj do	
4:        compute the eccentricity e(xi) for all xi∈Cj.	
5:        extract the cluster approximation graph GkCj.	
6:        compute the single linkage clustering hierarchy of GkCj	
7:        simplify the clustering hierarchy using a minimum branch size mb.	
8:        extract labels and probabilities for a ‘flat’ clustering.	
9:      end for	
10:     combine the cluster and branch labels and probabilities.	
11:     return the membership labels and probabilities.	
12:  end function	

The concepts behind density-based clustering can also be applied to detect branches within clusters by using an eccentricity measure in place of density, as shown in Fig. 2. Peaks in an eccentricity profile correspond to branches in the cluster, as shown in Figs. 2B and 2C. We define eccentricity as the distance to the cluster’s centroid:

(4) e(xi)=d(x¯Cj,xi),

where x¯Cj is the cluster’s membership-weighted average (Fig. 2A). This eccentricity measure can be computed in O(N). Comparable to density contour clusters, an eccentricity contour cluster is a maximal, connected subset of points with an eccentricity above some threshold {x|e(x)≥et}. As in functional persistence (Carlsson, 2014), eccentricity thresholds filter out cluster cores, which separates branches and makes them detectable as connected components.

Figure 2 Density-based clustering concepts behind FLASC.

(A) A within-cluster eccentricity e(xi) is defined for each point xi in cluster Cj based on distances to the cluster’s membership weighted average shown by the pentagon mark. (B) The cluster’s eccentricity profile visualised as contours on a height map. Peaks in the profile correspond to branches in the cluster. (C) Branches extracted from the cluster by FLASC indicated in colour. The cluster’s centre is given its own label. (D) The eccentricity contour tree describes how branches merge when considering lower eccentricity thresholds.

Like HDBSCAN*, we need connectivity between data points to determine whether two points are part of the same eccentricity contour cluster at a threshold et. In FLASC, we provide two solutions based on the cluster’s density scale in the form of cluster approximation graphs GkCj: the full approximation graph and the core approximation graph. Both graphs contain a vertex for each point in the cluster xi∈Cj but differ in which edges they include.

The full approximation graph adds all edges with dmreach(xi,xl)≤dmaxCj, where dmaxCj is the longest distance in the cluster’s minimum spanning tree (MST). The resulting graph accurately describes the connectivity within the cluster at the density where the last point joins the cluster. The space tree constructed by HDBSCAN* is used to retrieve these edges efficiently.

The core approximation graph adds all edges with dmreach(xi,xj)≤max{κ(xi),κ(xj)} to the cluster’s MST. The resulting graph accurately describes all connectivity represented by the MST. This graph can be seen as the cluster’s subgraph from the k-nearest neighbour graph over the entire data set. HDBSCAN* already extracted these edges when the core distances were computed, so this approach has a lower additional cost.

We can now recover the eccentricity contour tree as if it were a density contour tree by applying HDBSCAN*’s clustering steps to the cluster approximation graph with its edges weighted by min{e(xi),e(xl)}. This weighting ensures an edge has the eccentricity of the least eccentric point it connects. Specifically, we use the Union-Find data structure from McInnes & Healy (2017) to construct a single linkage dendrogram. The resulting hierarchy is simplified using a minimum branch size mb to recover the condensed branching hierarchy shown in Fig. 2D.

HDBSCAN*’s EOM and leaf strategies compute branch labels and membership probabilities from these condensed hierarchies. Points that enter the filtration after the selected branches have connected—i.e., points with the noise label—are given a single non-noise label representing the cluster’s centre. Finally, the cluster and branch labels are combined. By default, points in clusters with two or fewer branches are given a single label because two branches are expected in all clusters, indicating the outsides growing towards each other. The label sides as branches parameter can be used to turn off this behaviour and separate the ends of elongated clusters in the labelling. The cluster and branch probabilities are combined by taking their average value (Fig. 3A).

Figure 3 Different ways to combine cluster and branch membership probabilities.

The cluster and branch probability (A) average and (B) product are visualised with desaturation. (C) Points labelled by the geodesically closest branch root—i.e., the point closest to the branch’s weighted average—and desaturated as in (A). (D) Weighted branch membership for the orange branch is visualised by transparency. Branch memberships are computed from the traversal distance to the branch’s root.

Other labelling and probability combinations are possible. For example, the cluster and branch probability product more strongly emphasises the outsides of the branches (Fig. 3B). As in McInnes, Healy & Astels (2017), FLASC supports computing branch membership vectors that describe how strongly a point xi∈Cj belongs to each branch Bb⊂Cj. These membership values are based on the geodesic distances in the cluster approximation graph GkCj: dgeo(rBb,xi), where rBb is the branch’ root, i.e., the point closest to the branch’s membership-weighted average x¯Bb. The branch membership vectors can be used to label central points by the closest branch root, as in Fig. 3C. Alternatively, a softmax function can be used to convert dgeo(rBb,xi) into the membership probabilities:

(5) p(xi,Bb)=ecb(xi,Bb)/t∑Bl∈Cjecb(xi,Bl)/t,

where cb(xi,Bb)=1/dgeo(rBb,xi) and t is a temperature parameter (Fig. 3D).

Low persistent branches can be ignored using a branch selection persistence parameter, analogous to HDBSCAN*’s cluster selection epsilon parameter (Malzer & Baum, 2020). As branches do not necessarily start at zero centrality, branch selection persistence describes the minimum eccentricity range rather than a single eccentricity threshold value. The procedure that applies the threshold simplifies the condensed branch hierarchy until all leaves have a persistence larger than the threshold.

Stability

Stability is an important property of algorithms, indicating that their output differs only slightly when the input changes slightly. Two notions of stability are relevant for FLASC: (1) the algorithm has to provide similar results when run repeatedly on (different) samples of an underlying distribution, and (2) the detected branch hierarchies have to represent the clusters’ underlying topology accurately. The deterministic density-based design of FLASC provides stability in the first sense.

Vandaele et al. (2021) analysed the second notion of stability for graph-based branch detection, explaining that the graph approximation should accurately represent the underlying shape and the graph-based centrality function should accurately describe the points’ centrality in a cluster’s metric space (Cj,dmreach). For the normalised centrality used by Carlsson (2014), Vandaele et al. (2021) show that the bound on the bottleneck distances between true and empirical persistence diagrams is tight if the metric distortion induced by the graph and its maximum edge weight are small.

Both the full and core cluster approximation graphs used by FLASC satisfy the low maximum edge weight requirement, as their largest edge weight is the minimum mutual reachability distance required for all points in the cluster to be connected in the graph. Additionally, the metric distortion should be small as only edges in the local neighbourhood of data points are included because the clusters do not contain noise points.

Our eccentricity function (Eq. (4)) also meets Vandaele et al.’s requirement to be a c-Lipschitz-continuous function when considered over the cluster centrality graph’s edges:

(6) |max{e(xi),e(xj)}−max{e(xk),e(xl)}|≤cdmreach(xi,xl),

where c is a constant describing the continuity, (xi,xj)∈GkCj, (xk,xl)∈GkCj, and the mutual reachability between xi and xl is the largest of the four points. This property, however, does not guarantee that the detected hierarchy accurately represents the cluster’s topology because Eq. (4) is sensitive to an interplay between the cluster’s shape and the position of its centroid. Consider, for example, a U-shaped cluster. Topologically, this cluster is equivalent to an I-shaped cluster. Equation (4), however, will contain two local eccentricity maxima and three local eccentricity minima because the centroid is located between the U-shape’s arms. As a result, the detected branching hierarchy is indistinguishable from a Y-shaped cluster. We aim to show that the current approach strikes a good balance between computational cost and stability in the experiments presented in the next section.

Computational complexity

The algorithm’s most computationally expensive steps are constructing the full cluster approximation graphs and computing their single linkage hierarchies. Naively, the worst-case complexity for creating a cluster approximation graph is O(nc2), where nc is the number of points in the cluster. Usually, the average case is much better because the approximation graphs are rarely fully connected. After all, HDBSCAN*’s noise classification limits the density range within the clusters. Furthermore, the space tree re-used from the HDBSCAN* clustering step provides fast asymptotic performance for finding the graph’s edges. The exact run-time bounds depend on the data properties. They are challenging to describe (as explained in McInnes & Healy, 2017). However, an average complexity proportional to O(nelog⁡N) is expected, where ne is the number of edges in the approximation graph. Computing single linkage hierarchies from the cluster centrality graphs is possible in O(neα(ne)) using the Union-Find implementation from McInnes, Healy & Astels (2017) ( α is the inverse Ackermann function). Like McInnes & Healy (2017), we feel confident that FLASC achieves sub-quadratic complexity on average, which we demonstrate in the ‘Experiments’ section.

Experiments

This section presents two synthetic benchmarks and two exploration use cases on real-world data sets to demonstrate FLASC’s properties. The first benchmark compares FLASC’s branch detection ability to other clustering algorithms and demonstrates that FLASC provides similar output for different samples of the same underlying distribution. The second benchmark compares FLASC’s computation cost to other clustering algorithms to show that the branch-detection post-processing step is computationally cheap compared to the initial clustering step. Finally, the two exploration use cases demonstrate how detecting branch-based subgroups and branch hierarchies can help understand the structure of real-world datasets.

Branch detection ability

This first synthetic benchmark compares the branch detection ability of several clustering algorithms. The benchmark is designed to answer the following research question: how well can FLASC detect branches compared to other clustering algorithms?

Six algorithms with Python implementation were selected from different clustering algorithm categories: HDBSCAN* (Campello, Moulavi & Sander, 2013; McInnes & Healy, 2017) represents density-based clustering algorithms and serves as a baseline; Single Linkage Clustering (SLINK) (Sibson, 1973) represents hierarchical clustering algorithms; kMeans (MacQueen, 1967; Lloyd, 1982; Arthur & Vassilvitskii, 2006) represents partitional clustering algorithms; Spectral clustering (Damle, Minden & Ying, 2019) represents graph-based clustering algorithms; and CDC (Peng et al., 2022) represents boundary-seeking clustering algorithms4 .

We expect that kMeans, Spectral clustering, CDC, and FLASC will be able to detect branching structures to varying degrees. In contrast, we expect SLINK and HDBSCAN* will struggle due to the lack of low-density regions separating the branches from the clusters. The following two subsections explain how the data for this benchmark was generated and how we measured the algorithms’ branch detection accuracies, respectively.

Datasets

The datasets for this benchmark contain four star-shaped clusters and uniformly distributed noise points (Fig. 4). Each star is parameterised by its number of branches ( 3, 4, 5, 10), branch lengths ( 1.8, 2.3, 2.0, 3.5), noise levels ( 0.2, 0.2, 0.02, 0.1), and points per branch ( 40, 150, 20, 100). The stars were generated and positioned in 2D. Branches were evenly spread and contain points exponentially spaced from the centre outwards, ensuring density is highest at the stars’ centres and lowest at the branch ends. This configuration prevents the branches from reliably containing local density maxima. Normally distributed noise ( μ=0) was added to the points in 2, 8, or 16 dimensions. The noise level was converted to a standard deviation σ=nr1/d to correct for the number of dimensions d. Uniformly distributed noise points were added to simulate outliers. The number of uniform noise points was 5% of the clean data set size. All structure is present in the first two dimensions. The additional dimensions contain only noise.

Figure 4 Clusters found by the algorithms using optimal parameter values (Table 1) in (A) 2D, (B) 8D, and (C) 16D.

Note that all structure is contained in the first two dimensions, the other dimensions contain only noise. Colour indicates the detected clusters. Hues are repeated when more than 20 clusters are detected, so points with the same colour in different parts of the data can represent different clusters. The CDC implementation only supports 2D data.

Table 1 Optimal parameter values per algorithm as found by the grid search described in the ‘Evaluation and settings’ sub-section.

Algorithm	Parameter	Dimensions	
		2	8	16	
FLASC	Approximation graph	Core	Core	Core	
Cluster selection method	Leaf	Leaf	Leaf	
Min. samples ( k)	6	6	2	
Min. cluster size ( mc)	70	100	84	
Min. branch size ( mb)	10	8	14	
HDBSCAN*	Cluster selection method	EOM	EOM	Leaf	
Min. samples ( k)	2	20	2	
Min. cluster size ( mc)	20	20	58	
CDC	Num. neighbours ( k)	2	N/A	N/A	
Ratio	0.61	N/A	N/A	
kMeans	Num. clusters	27	27	27	
SLINK	Num. clusters	27	27	27	
Spectral	Num. clusters	27	23	18	
Note:

These parameter values were used to compare the algorithms in this benchmark. The CDC implementation only supports 2D data.

Evaluation and settings

Five data sets were sampled in 2D, 8D, and 16D to evaluate the algorithms. A grid search was used to find optimal parameter values for each algorithm, ensuring they are compared on their best performance. For SLINK, Spectral clustering, and kMeans, the number of clusters was evaluated between 4 and 27 in six linear steps, with 23 being the true number of subgroups in the data. Spectral clustering was evaluated with QR factorisation to extract clusters from the eigenvalues (Damle, Minden & Ying, 2019). For CDC, the ratio parameter was evaluated between 0.5 and 0.95 in five linear steps. CDC’s num. neighbours and FLASC and HDBSCAN’s min. samples were evaluated between 2 and 20 in six linear steps. FLASC and HDBSCAN’s min. cluster size was evaluated between 20 and 100 in ten exponentially spaced steps. FLASC’s min. branch size was evaluated between 3 and 50 in ten linear steps. FLASC and HDBSCAN’s cluster selection method was evaluated with both the EOM and leaf strategies. FLASC’s branch selection method was set to the value of cluster selection method. Finally, FLASC was evaluated with the full and core cluster approximation graphs.

The resulting data point labels were stored for each evaluation and used to compute the Adjusted Rand Index (ARI) (Hubert & Arabie, 1985). ARI values describe the agreement between ground truth and assigned labels adjusted for chance. We selected the parameter values that maximised the average ARI over all five data sets. Table 1 shows the selected optimal parameter values for each algorithm.

Results

Figure 4 shows which clusters the algorithms detected with their optimal parameters (Table 1). The figure also lists the average ARI scores over the five generated data samples. Only FLASC achieved ARI scores greater than 0.61 and assigned each branch to a distinct group. The other algorithms achieved ARI scores lower than 0.36. SLINK and HDBSCAN* assigned whole stars to single groups, thereby not detecting most branches. CDC, kMeans, and Spectral clustering did assign multiple groups to the stars, but these groups did not correspond to the branches. Some groups spanned multiple branches, and some branches were given multiple groups. Generally, these findings agree with our expectation that clustering algorithms struggle to detect branches as clusters and highlight the benefit FLASC brings for detecting this type of pattern.

Next, we explore FLASC’s parameter sensitivity using results from the 2D grid search. Figure 5 shows the average ARI curves for the min. samples ( k), min. cluster size ( mc), and min. branch size ( mb) parameters. The curves do not show the optimal performance that can be reached by combining the best parameter values—as indicated by the black crosses—because the averages are computed over five data samples and the evaluated (non-optimal) values for the other parameters. Instead, the curves summarise how performance changes by varying the parameters, indicating the algorithm’s sensitivity. Shaded areas around the curves indicate a 95% confidence interval around means.

Figure 5 Average ARI curves for FLASC’s main parameters as achieved in the 2D parameter grid search: (A) min. samples, (B) min. cluster size, and (C) min. branch size.

The ARI averages were computed over five repeated runs and the evaluated (non-optimal) values for the other parameters. As a result, the averages do reach the optimal performance indicated by the black crosses. Shaded areas around the curves indicate 95% confidence intervals around the means. The legend indicates the colour and stroke used to encode the full and core approximation graphs and EOM and leaf cluster selection strategies.

The average ARI curve for min. samples is shown in Fig. 5A. The curve contains a peak around k=10, slightly higher than the optimal value of k=6. This pattern indicates that too large values of k reduce the algorithm’s performance. Generally, k should be set so that the k-closest neighbours of points within clusters are noticeably closer than the k-closest neighbours of noisy points between clusters. Higher values reduce variation and peaks in the modelled density profile, leading to larger clusters. In addition, higher values increase connectivity in the cluster approximation graphs, which reduces the algorithm’s sensitivity to branches. As a result, we tend to set k between 2 and 25.

The curves for min. cluster size and min. branch size are shown in Figs. 5B and 5C, respectively. For these parameters, the averages were computed with k=6. Both parameters show stable performance once they reach a large enough value to exclude noisy clusters ( mc>50) and branches ( mb>3). Similarly, both parameters should not be set too large, as that would exclude true clusters and branches ( mb<37). Generally, these parameters should be set to the smallest sizes one is interested in finding. The exact sizes vary on a case-by-case basis and may require some domain knowledge to deduce. When larger thresholds are used, the cluster and branching hierarchies contain fewer and larger leaf clusters. So, these parameters can fine-tune clustering results when using the leaf selection strategy.

The different cluster approximation graphs and cluster selection methods have similar curves in Fig. 5. We provide the following guidelines for tuning these parameters. The core cluster approximation graph is appropriate when clusters span large density ranges or contain branching structures at multiple densities. The full approximation graph is more suitable for finding branches at the lowest density within a cluster. The leaf selection strategy works well for finding all density peaks in the data. It tends to find many smaller clusters and can indicate many points as noise. The EOM strategy is more suitable for finding larger clusters containing multiple density peaks. The additional selection parameters can filter out low-persistent clusters and branches when needed.

Finally, we explore FLASC’s practical stability in the sense of its output similarity on the five sampled datasets. The output similarity is explored through unweighted average positions of the detected groups. We refer to these positions as the groups’ centroids. For each ground-truth centroid, we selected the closest centroid FLASC detected with its optimal parameters (Table 1). The centroids are shown in Fig. 6, with the black crosses indicating the ground-truth centroids and the coloured dots indicating their closest detected centroid. The 95 percentile distance to the closest detected centroid is shown as a line to indicate the centroids’ spread and describe the variation in FLASC’s output. Generally, the detected centroids are located close to the ground-truth centroids, and the 95 percentile distances are considerably smaller than the longest branches.

Figure 6 Output similarity of FLASC with optimal parameters (Table 1) on the five data samples in (A) 2D, (B) 8D, and (C) 16D.

Unweighted average positions (centroids) for the ground truth groups are indicated by black crosses. The closest centroids FLASC detected are drawn as coloured dots for each ground truth group. The 95 percentile distance between the ground truth centroids and their closest detected centroids is shown as a line in an annotation in the bottom right of each plot. The lines indicate that the detected centroids’ spread is considerably smaller than the largest branch lengths.

Computational performance

Now, we turn to the computational performance. This second synthetic benchmark is designed to answer the following research question: how does FLASC’s compute cost compare to other clustering algorithms?

The algorithms selected in the previous benchmark are compared on their run time scaling over the data set size and number of dimensions. Given the challenges in accurately benchmarking computational performance (Kriegel, Schubert & Zimek, 2017), we limit this comparison to the trends in run time scaling over data set size and number of dimensions for specific implementations. The following two subsections explain how the data for this benchmark was generated and how we measured the algorithms’ compute cost.

Datasets

A Gaussian random walk process was used to generate data sets that contain non-trivially varying densities and branching structures in a controlled environment. For a space with d dimensions, c uniform random starting points were sampled in a volume that fits five times the number of to-be-generated clusters. Then, five 50-step random walks were sampled from each starting point. Every step moved along one of the dimensions with a length sampled from a normal distribution ( μ=0, σ=0.1). The resulting point clouds have more varied properties than the Gaussian blobs used in a run-time comparison of HDBSCAN* (McInnes & Healy, 2017). Note that the number of (density-based) clusters in each point cloud may differ from the number of starting points c due to possible overlaps or sparse regions in the random walks. This data generation process also ensures that structure is present in all d dimensions, reducing the space trees’ effectiveness in accelerating HDBSCAN* and FLASC. Consequently, the detected scaling trends do not overestimate the algorithms’ performance.

Evaluation and settings

The random walk data sets were generated with varying numbers of dimensions ( 2, 8, 16) and starting points ( 2 to 800 in 10 exponentially spaced steps). Ten data sets were sampled for each parameter combination, resulting in 300 point clouds.

The algorithms were compared using Python implementations: HDBSCAN* version 0.8.405 (McInnes, Healy & Astels, 2017); kMeans, SLINK and Spectral clustering from Scikit-Learn version 1.5.2 (Pedregosa et al., 2011); and FLASC version 0.1.36 . The CDC implementation is included in FLASC’s repository.

HDBSCAN* and FLASC were evaluated with min. samples k=5, min. cluster size mc=100, min. branch size mb=20. These parameter values allow the algorithms to find clusters and branches that are slightly smaller than how they are generated. kMeans, SLINK, and Spectral clustering were evaluated with k=c, effectively attempting to recover one cluster for each starting point. CDC was configured to use k=10 and a ratio of 0.1. Multiprocessing support was turned off for all algorithms to better describe the algorithms’ intrinsic complexity.

Time measurements were performed with a 5.4 GHz AMD R7 7700X processor. Each algorithm was evaluated on each data set once, recording the run time and number of detected clusters. The smallest data set for which an algorithm required more than 30 s was recorded for each number of dimensions. Larger data sets were not evaluated for those algorithms.

Results

Figure 7 shows the measured run times in seconds over the data set size and number of dimensions. Quadratic regression lines are drawn to visualise the algorithms’ scaling behaviour. The shaded areas around each line indicate the regression’s 95% confidence interval.

Figure 7 Benchmark run times (s) over the data set size and number of dimensions: (A) 2D, (B) 8D, and (C) 16D.

The algorithms’ scaling trends are visualised by quadratic regression lines relating compute time to the number of points. The shaded areas around each line indicate the regression’s 95% confidence interval.

There are several patterns of note in Fig. 7. CDC and Spectral clustering show the steepest quadratic scaling trends in all evaluated dimensions. SLINK exhibits a shallower quadratic scaling trend. kMeans achieved the shallowest scaling trend in all three conditions. HDBSCAN* and both FLASC variants regression lines fall between SLINK and kMeans. In the two dimensional case, their trends are most similar to kMeans. As more dimensions are added, their trends approach SLINK’s trend. This pattern is expected because this dataset is designed to challenge the algorithms’ use of space trees to accelerate the clustering process.

The difference in compute time between the two FLASC variants and HDBSCAN* is small compared to the total compute time, indicating that the branch detection post-processing step is relatively cheap compared to the initial HDBSCAN* clustering step, making it a viable option in practical data analyses workflows.

Use case: diabetes types

Next, we present a data exploration case in which identifying branch-based subgroups is essential to understand the data. Reaven & Miller (1979) attempted to clarify a “horseshoe”-shaped relation between glucose levels and insulin responses in diabetes patients. Three of the metabolic variables they measured were very informative in a 3D scatterplot, showing a dense core with two less-dense branches, which they considered unlikely to be a single population. Seeing that plot was instrumental in their understanding of the data (Miller, 1985).

More recently, Singh, Memoli & Carlsson (2007) used the same data set to demonstrate how Mapper visualises these flares without manually specifying which dimensions to plot. Their analysis leveraged the flares’ lower density, allowing them to be detected with a density-based lens function instead of a centrality measure. In general, though, local density minima do not always relate to branches, especially for data sets with multiple branching clusters.

In this use case, we show how FLASC can detect the branching pattern in this data set and classify the observations by their branch without manually extracting the flares from a visualisation.

Evaluation and settings

The data set—obtained from Andrews & Herzberg (1985)—contains five variables describing 145 subjects: the relative weight, the plasma glucose level after a period of fasting, the steady-state plasma glucose response (SSPG), and two areas under a curve—one for glucose (AUGC) and one for insulin (AUIC)—representing the total amount observed during the experimental procedure described in Reaven & Miller (1979). All five variables were z-score normalised and used to compute the Euclidean distance between subjects.

Both FLASC and HDBSCAN* were evaluated on the normalised data set. FLASC was tuned to find a single cluster with multiple branches by setting min. samples k=5, min. cluster size mc=100, min. branch size mb=5, and enabling allow single cluster. HDBSCAN* was tuned to find multiple clusters with min. samples k=5 and min. cluster size mc=10.

Results

Figure 8 shows the detected subgroups encoded using colour on the 3D scatterplot. FLASC’s classification distinguishes the branches from the central core (Fig. 8A). The algorithm also finds a low-persistent flare representing the central core’s bottom. This flare could be ignored by specifying a persistence threshold. In contrast, HDBSCAN*’s classification does not find the branches (Fig. 8B). Instead, it finds part of the left branch as a small low-persistent cluster and merges most of the right branch with the central core.

Figure 8 Subgroups detected by (A) FLASC and (B) HDBSCAN* shown in a 3D scatterplot over the area under the plasma glucose curve (AUGC), the area under the insulin curve (AUIC), and the steady state plasma glucose response (SSPG) from Reaven & Miller (1979).

Grey points were classified as noise.

This case study demonstrated how FLASC detects branch-based subgroups that do not contain local density maxima without having to specify the relevant features in advance or extract the subgroups visually. Practically, FLASC would have made it easier for researchers to detect the three groups in this data set, which was relevant for understanding diabetes and its causes.

Use case: cell development

Finally, we demonstrate a use case where detecting a branch hierarchy is important for understanding the data set. Specifically, we analyse a cell development atlas for the C. Elegans—a small roundworm often used in biological studies—by Packer et al. (2019). They analysed gene expressions in C. Elegans embryos to uncover the trajectories along which cells develop. Broadly speaking, this data set describes what happens in cells as they develop from a single egg cell into all the different tissues within fully grown C. Elegans worms.

After pre-processing, the data set appears to contain clusters and branching structures when viewed in a 3D projection (Packer et al., 2019). In this use case, we demonstrate that FLASC’s branch hierarchy provides interesting information about the data set’s structure even when the main subgroups can be detected as clusters.

Evaluation and settings

The data and pre-processing scripts were obtained from Monocle 3’s (Cao et al., 2019) documentation (Pliner, Kouhei & Trapnell, 2022). The pre-processing stages normalise the data, extract the 50 strongest PCA components, and correct for batch effects using algorithms from Haghverdi et al. (2018).

HDBSCAN* was evaluated on the pre-processed data with an angular distance metric because its optimised code path does not support the cosine distance metric. HDBSCAN* was tuned to find multiple clusters with min. samples k=5 and min. cluster size mc=50.

FLASC was evaluated on a 3D UMAP (McInnes, Healy & Melville, 2020) projection that denoised the data set. Using three instead of two dimensions reduces the chance of branch overlaps in the embedding. UMAP used the angular distance metric to find 30 nearest neighbours. The disconnection distance parameter was set to exclude the 16-percentile least dense points detected by HDBSCAN*, thereby preventing shortcuts across the data’s structure. The largest connected component in the resulting UMAP graph was projected to 3D while varying the repulsion strength to avoid crossings and ensure connected structures remained close. The same layout procedure was used to project the graph to 2D to visualise the data.

FLASC was tuned to detect the branching hierarchy of the dataset as a single cluster by selecting min. samples k=3, min. cluster size mc=500, min. branch size mb=50, enabling allow single cluster. Branches were detected using the core cluster approximation graph and selected using the leaf strategy.

Results

Figures 9A and 9B show 2D UMAP projections of the data. Data points are coloured to indicate the detected clusters and branches, respectively. There are two main differences between the branches and the clusters. Firstly, two regions where branches merge are detected as clusters, namely clusters 10 and 14. These regions do not have a distinct branch label but are identifiable in the branch hierarchy (Fig. 9D). Secondly, only branch 13 was not detected as a cluster in this dataset, indicating that the branches represent regions with high local density. Considering that the branches correspond to developmental end-states, it is unsurprising that local density maxima occur within them. One could imagine that the variation in gene expression is higher during development and that fully developed cells are observed more frequently. Both scenarios could cause these local density maxima.

Figure 9 Results for the single cell gene expression use case using 50 dimensional pre-processed data from Packer et al. (2019).

(A) 2D UMAP projection (McInnes, Healy & Melville, 2020) coloured by HDBSCAN* clusters detected in the pre-processed data. (B) The same projection coloured by FLASC branches detected from a 3D UMAP projection. (C) and (D) show cluster and branch hierarchies, respectively. The icicle plots were adapted from McInnes, Healy & Astels (2017) to indicate selected clusters with colour and labels.

More interesting are the differences between the cluster and branch hierarchies. Figures 9C and 9D visualise these hierarchies as an icicle plot. These designs are adapted from McInnes, Healy & Astels (2017) to indicate the selected branches and clusters using colour and a label. Segment widths encode the number of points in the tree below the segment. The hierarchies highlight that while HDBSCAN* detects the branches as clusters, it does not capture the trajectories. For example, the hierarchy does not reflect that clusters 0 and 1 connect to the whole data set through cluster 10. FLASC’s branch hierarchy, on the other hand, more closely resembles the data’s shape. For example, the hierarchy describes the embedding’s left and right sides with five and six branches, respectively. Generally, branches that merge into the cluster near each other are also close in the branch-condensed tree. The branch-condensed tree captures only the most eccentric connection for branches connected to multiple other branches in the cluster approximation graph.

This use case demonstrated that FLASC’s branch hierarchy provides information about the data’s shape that may not be obvious from cluster hierarchies. In addition, we found it beneficial to suppress noisy connectivity using dimensionality reduction techniques when detecting branches.

Discussion

Two synthetic benchmarks and two real-world use cases were performed to demonstrate FLASC and its properties. We start our discussion by providing remarks for each benchmark and use case.

The first benchmark compared FLASC to other clustering algorithms in their ability to detect branches that do not contain a density maximum. The benchmark quantified performance using the Adjusted Rand Index (ARI) (Hubert & Arabie, 1985) and explored FLASC’s output similarity on samples of the same underlying distribution. Of the evaluated algorithms, only FLASC reliably detected the branches, indicated by ARI scores greater than 0.60. Spectral clustering and kMeans were the only other algorithms to reach ARI scores above 0.30. However, these algorithms have practical limitations for exploratory data analysis workflows. For example, both require specifying the number of clusters a priori, which is challenging for unfamiliar data. In addition, spectral clustering is computationally expensive, and kMeans can produce different results on repeated runs. FLASC does not have these limitations, as demonstrated by the centroid spread (Fig. 6) and the second benchmark.

This first benchmark could be expanded to investigate how well FLASC deals with unequal branch lengths. The weighted average data point—and eccentricity measure as a result—may not accurately reflect the centre of such clusters. Consequently, FLASC’s branch hierarchy will be a less accurate representation of the underlying topology but should still detect the branches. Monocle 3 (Qiu et al., 2017) deals with this problem by manually selecting the centre point in a 2D projection (Pliner, Kouhei & Trapnell, 2022). Other eccentricity measures discussed below could also improve FLASC’s performance in such cases.

The computational performance benchmark demonstrated that FLASC’s computational cost scales similarly to HDBSCAN*. Additionally, their scaling trends become more similar with more dimensions. kMeans provides even quicker run times but is limited in usability due to its lower stability and predefined number of clusters. The other evaluated algorithms were not competitive in terms of run time. The benchmark provides a pessimistic view of FLASC and HDBSCAN* for practical use because multiprocessing was turned off, and the data was designed to be challenging for the space trees that accelerate these algorithms. On the other hand, extracting the full cluster approximation graph can be more expensive than reported for clusters that span larger density ranges.

The diabetes types use case demonstrated a real-world dataset in which branches that are not detectable as density-based clusters represent meaningful groups in the data. FLASC is designed to detect such branches without knowing they exist a priori or manually extracting them from a visualisation. The cell development use case showed how FLASC behaves on a more complex data set. Here, the groups were detectable by both FLASC and HDBSCAN*. FLASC still benefits exploration because its branch hierarchy describes the data’s shape. Structure learning algorithms, as described in Section ‘Related Work—Structure Learning’, can provide even more information about the data’s shape at more computational costs.

FLASC’s practical value

As demonstrated by the cell development use case, the argument that branches are not detectable as clusters only applies when they do not contain local density maxima. Subpopulations tend to have some location in feature space where observations are more likely. These locations are detectable as local density maxima, allowing data points surrounding them to be classified as a particular cluster. If one is only interested in the existence of subgroups, then FLASC only provides a benefit on datasets where relevant subgroups are sparse (e.g., the diabetes types use case). If one is also interested in the clusters’ shapes, then FLASC’s branch hierarchy provides information that cannot be extracted from a cluster hierarchy. We envision FLASC as a valuable tool for exploring unfamiliar data, providing guidance into which subpopulations exist and informing follow-up questions. Knowing that a cluster may represent multiple subgroups can be very relevant.

Alternative eccentricity metrics

The presented FLASC algorithm uses a geometric distance-to-centroid metric to describe how eccentric data points are within a cluster (Fig. 2A, Eq. (4)). An interesting alternative is an unweighted geodesic eccentricity, which measures path lengths between each data point and the cluster’s root point in the cluster approximation graph. Here, the root point can be chosen as the data point closest to the cluster’s centroid, as we did for the branch-membership vectors (Fig. 3). Such a geodesic eccentricity would agree with the notion that distances in high dimensional data may not accurately reflect distances along the intrinsic structure of a data set, which was one of the motivations for Reversed Graph Embeddings (Mao et al., 2017). It would also be closer to the maximum shortest-path centrality metric used by Vandaele et al. (2021).

Several trade-offs between the geometric and geodesic eccentricity metrics made us choose the geometric one: Computing the geodesic eccentricity is more expensive because it requires an additional traversal over the entire cluster approximation graph. The extra cost, however, should be low compared to other parts of the algorithm.

The resolution of the unweighted geodesic eccentricity is lower, as it expresses the number of edges to the root point. As a result, zero-persistent branches are more likely to occur. In addition, it reduces the detectability of small branches that are well connected. On the other hand, that can be seen as beneficial noise suppression. In addition, the branch selection persistence parameter becomes more interpretable and would represent the traversal depth of a branch in the approximation graph.

The cluster’s centroid may lie outside the cluster, resulting in a root point and eccentricity values that do not accurately describe its centre. For example, imagine a U-shaped cluster. The centroid would lie in between the two arms, and the root would lie in one of the arms. As a result, the geodesic metric would find one smaller and one larger branch rather than two equal branches. On the other hand, the geometric eccentricity finds the two branches and the connecting bend as three separate groups. Confusingly, it also contains two regions with a local eccentricity maximum, which FLASC gives a single label. Placing the root at an arbitrary eccentric location, as in Ge et al. (2011), avoids this issue but necessitates a different interpretation of the branch hierarchy and membership probability.

FLASC’s general process can also be used with measures that capture other aspects than eccentricity. At its core, FLASC consists of two filtrations, one to determine the connectivity between data points and one to analyse a signal on the resulting graph. The process would then describe how many distinct local minima (or maxima) of the measure exist within the clusters. The resulting interpretation does not have to relate to the cluster’s shape. For example, one could use the boundary coefficient (Vandaele, Saeys & De Bie, 2020) or direction centrality metric (Peng et al., 2022) to create an efficient and principled boundary-seeking clustering algorithm.

One could even interpret FLASC as two applications of HDBSCAN*: one over the density and one over the eccentricity. This perspective raises a possible improvement to the algorithm by translating the mutual reachability concept to the centrality metric. The idea of ‘pushing away points in low-density regions’ can also be applied to the centrality, emphasising the centrality difference between the centre and branch ends. Additionally, smoothing the centrality profile by incorporating neighbouring values could improve the algorithm’s robustness to noise. The additional computational cost should be low, as points’ neighbours are already known when the centrality is computed. Another way to improve noise robustness could be to implement the mutual k-nearest neighbour approach Dalmia & Sia (2021) used to improve cluster separation in UMAP projections. It would provide a subgraph of the core approximation graph that better reflects the cluster’s connectivity in high dimensional data sets. We leave evaluating these ideas for future work.

Visually summarising data’s shape

A strength of Mapper (Singh, Memoli & Carlsson, 2007) and Reversed Graph Embeddings (Mao et al., 2017) is that they can summarise the data’s shape using intuitive visualisations. While FLASC’s branch-condensed tree provides some information about the clusters’ shapes, interpreting the shape is not trivial. Studying how well two-dimensional layouts of FLASC’s cluster approximation graphs work as shape summarising visualisations would be an interesting future research direction. These graphs directly encode the connectivity used by the algorithm. Another benefit is that—unlike in Mapper—all (non-noise) data points are represented in the graphs once. Directly visualising the graphs, however, probably does not scale to larger sizes in terms of computational cost for the layout algorithm and visual interpretability. Ways to summarise the networks would have to be found, which could be based on kMeans centroids like in Reversed Graph Embeddings (Mao et al., 2017), local density maxima in the cluster, or a Reeb-Graph approach similar to Ge et al. (2011).

Conclusion

We presented the FLASC algorithm that combines HDBSCAN* clustering with a branch-detection post-processing step. We have shown that the algorithm can detect branch-based subgroups that do not contain local density maxima in real-world data without specifying features of interest or manually extracting the branches from a visualisation. In addition, we demonstrated that branching hierarchies found by FLASC can provide information about the data’s shape that is not present in HDBSCAN*’s cluster hierarchy. Two synthetic benchmarks demonstrated FLASC’s branch detection ability and indicated FLASC’s computational performance scales similarly to HDBSCAN*.

We thank Kris Luyten for his comments on an early version of the manuscript. Grammarly was used in the preparation of this manuscript.

Additional Information and Declarations

Competing Interests

The authors declare that they have no competing interests.

Author Contributions

Daniël M. Bot conceived and designed the experiments, performed the experiments, analyzed the data, performed the computation work, prepared figures and/or tables, authored or reviewed drafts of the article, and approved the final draft.

Jannes Peeters conceived and designed the experiments, authored or reviewed drafts of the article, and approved the final draft.

Jori Liesenborgs conceived and designed the experiments, authored or reviewed drafts of the article, and approved the final draft.

Jan Aerts conceived and designed the experiments, authored or reviewed drafts of the article, and approved the final draft.

Data Availability

The following information was supplied regarding data availability:

The code and data used in this work are available at Zenodo: Bot, J., Peeters, J., Liesenborgs, J., & Aerts, J. (2025). Code & Data for “FLASC: A Flare-Sensitive Clustering Algorithm”. (0.1.4). Zenodo. https://doi.org/10.5281/zenodo.14888003.

4 There was no ready-to-use Python implementation for CDC at the time of testing. However, the publicly available code from Peng et al. (2022), while not optimised, was easy to integrate in our benchmark.

5 HDBSCAN* was evaluated at commit 2fada32 containing changes to be included in version 0.8.41.

6 FLASC was evaluated at commit 9e9cb13 containing changes to be included in version 0.1.4.

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
