# Peer review of "FLASC: a flare-sensitive clustering algorithm"

_PeerJ Computer Science, doi:10.7717/peerj-cs.2792_

## Round 0.1 · original submission · Major Revisions

While the reviewers agree that the paper has potential to be published, there is a consensus that it needs major revision, though the various reviewers have diverse set of individual reasons. Please pay attention to all those constructive feedbacks (even as I emphasize some of the most crucial ones).
- Provide a proper, well-structured, and up to date literature survey. Use that to identify the gaps in current state of the art, and identify which gap(s) are being addressed in your work.
- Provide proper validation of your results both in terms of its performance in quality of the algorithm output, as well as other practicalities like time/space complexity (analytical results validated with experiments).
- Your validation should of-course include diverse inputs where the algorithm performs favourably w.r.to state of the art, but you should also expose its shortcomings with experiments and identifying scenarios where it might not. So to say, provide some 'stress tests'.

Please understand that a major revision decision does not necessarily suggest a pathway to an acceptance, as such, do not rush the revision. You may ask for extensions to revise so that you can address well all the issues. Your priority should be to ensure that the quality of revision is good, even if that means you need more time to do so.

Reviewer 1 ·

Basic reporting

This paper was written very well and has a good opportunity to be published in this great journal, after addressing the following concerns in the first round:

- I encourage you to add more detail about your core contributions in the abstract. Abstract has five-section and you should follow the best practices in your area! Please also mention the novelties in the abstract.
- Long paragraphs.
- Please bring some facts and figures in the introduction to support the ideas.

Experimental design

- Literature review is very short and old! We are in jan 2024! You have not covered the knowledge edge! Please clarify the contribution of the paper according to the research gap. Many recent papers in the area can be added to the literature review. You have not referred to the main works in this area. I do not propose you any special reference due to ethical issues. The authors should review the recent works in these areas. I simply search and find following works and similar algorithm like, TGA, SEO, RDA, HBA, FHO, etc. Do and implement your algorithm in professional way by needed experiments... Also, you should cover these algorithms in the LR and refer to the works as samples for their usages.

Also, you should cover these algorithms in the LR and refer to the works as samples for their usages.
HBA, KA, SEO, FHO, TGA, etc.

Validity of the findings

- Please add research gap section.


I go with a major revision in this step and waiting for your corrections. Then, I give you my technical comments.

Additional comments

- Check the English presentation of this paper to remove the typos mistakes.
- Findings, limitations, and recommendations of this paper can be discussed more in the conclusion section.
- Please bring and focus on future research directions.

Cite this review as
Anonymous Reviewer (2025) Peer Review #1 of "FLASC: a flare-sensitive clustering algorithm (v0.1)". PeerJ Computer Science

Reviewer 2 ·

Basic reporting

* There are certain terms that are not clear (at least to this reviewer) in the following lines:

- 14: "their shape can represent", please be more explicit in the paper what a shape is, provide a definition or an intuition of it.

- 17: in how far do branches 'describe' within-cluster connectivity? please elaborate on that within the paper

- 19: 'stable outputs', stable in which term/with respect to what? Please elaborate on that

- 20: 'flare sensitive' what is meant by that term? please define or provide intuitions for that within the text

- 25: 'traditional algorithms', what is understood by the authors via the term 'traditional'?

- 31: 'subpopulations': what is meant by that term? do you mean subsets of objects/data points?

- 36: "Flares are connected in the simplicial complex", please provide a definition of flares in context of topological data analysis (TDA)

- 55+56: "branching hierarchies" and "branch-based subgroups of clusters": Please provide a definition of these terms. Furthermore what is the difference in context of your manuscript regarding the terms subgroups and subpopulations?

- 62+63: good! nice reference of existing work for branch detection in 3d models of plants

- 70: "...intuitive ... and branch sizes..." in how far are branch sizes intuitive? please elaborate

- The elaboration on HDBSCAN* is very well done and elaborate!

Experimental design

- 155: "distance between them is smaller than or equal to 1/lambda_t", why is that the case or thresholded by that? please elaborate on that matter

- 178: "no cluster selection epsilon", what is meant by that? do you refer to an epsilon-neighborhood radius? please elaborate

-236: "The eccentricity function (Equation (4)) is more complex to analyse..." with respect to what? what makes it so complex? please elaborate

- You compare FLASC against k-means and HDBSCAN*, while the latter makes sense, since your proposed method shares many theoretical and algorithmic foundations, it remains unclear why you do not compare against other methods. Most important this reviewer recommends to compare also against Hierarchical Clustering (e.g. SLINK) and Spectral clustering. Both of them rely on trees/graphs as their underlying structures. Prior works by Hess et. al [The relationship of DBSCAN to matrix factorization and spectral clustering] and Beer et. al [Connecting the Dots--Density-Connectivity Distance unifies DBSCAN, k-Center and Spectral Clustering], also revealed commonalities between DBSCAN, Spectral Clustering.

Furthermore regarding the discovery of branches, this reviewer recommends to compare FLASC against 4C, a density-based clustering algorithm [Computing Clusters of Correlation Connected Objects] that is capable to detect linear correlations (to a certain degree discovering linear branches) within a given dataset.

To test your method against the proposed algorithms has the purpose to substantially underline the power of your method by comparing against other methods that should, in theory, be capable to detect it.

- while the figures with the ARI scores are convincing (cf. Fig. 4), a question that comes up is: how does FLASC behave if the shape of three branches is more like a Y-shape or in 3D two Y-shapes with a certain viccinity to each other. Please provide 2-3 more complex datasets in which FLASC is challenged to convince the readers that it is also capable to deal with different shapes and different proximities of clusters to each other

Validity of the findings

The used methods are well established and based on the datasets on which the experiments have been conducted they seem meaningful, however, as mentioned in 2. Experimental Design, more experiments are needed to investigate the capabilities and limitations of that method.

Cite this review as
Anonymous Reviewer (2025) Peer Review #2 of "FLASC: a flare-sensitive clustering algorithm (v0.1)". PeerJ Computer Science

Reviewer 3 ·

Basic reporting

1. The selection criteria of the algorithm parameters (e.g., distance threshold, minimum branch size, etc.) are not discussed in detail during the experiment. It is suggested to add the comparison experiment of the performance of the algorithm under different parameters to show the robustness and adaptability of the FLASC algorithm.

2. In the past three years, several studies have addressed the challenge of flare detection. Notable examples include, “Clustering by Measuring Local Direction Centrality for Data with Heterogeneous Density and Weak Connectivity” (Nature Communications), and TWStream (IEEE TFS). These algorithms merit further discussion.

Experimental design

1.The experimental comparison object focuses on a major density clustering algorithm (HDBSCAN*). In order to more fully demonstrate the superiority of FLASC, it is suggested to introduce other density clustering algorithms, such as, Block-Diagonal Guided DBSCAN Clustering(IEEE TKDE github: https://github.com/y66y/TKDE) and M3W (IEEE TNNLS github: https://github.com/Du-Team/M3W).

2.The current experimental datasets are mainly simulated data or datasets of specific nature, and lack the testing of real application scenarios. It is suggested to add some datasets from real applications to verify the practicality of FLASC.

Validity of the findings

The time complexity of FLASC algorithm is theoretically analyzed in the paper, so the time comparison experiments are very much expected.

Cite this review as
Anonymous Reviewer (2025) Peer Review #3 of "FLASC: a flare-sensitive clustering algorithm (v0.1)". PeerJ Computer Science

·

Basic reporting

No comment

Experimental design

1. Why was the FLASC algorithm compared with the K-means algorithm, if it is known by the clustering community that this algorithm does not detect groups of irregular figures? I think that being very popular is not a good argument for the comparison.
2. What are the values of the parameters used by the DBSCAN* algorithm.
3. How was the optimal number of groups determined in the K-means algorithm?
4. It is recommended to compare the proposed algorithm with an OPTICS algorithm.
5. Could you explain and give arguments for the “Evaluation and settings” sections of all the cases presented.
6. Could you explain what is written in lines 402-405, why K=5, min cluster size Mc = 100…..
7. How was the efficiency of the clusters obtained by the three algorithms measured?

Validity of the findings

No comment

Cite this review as
Rendón lara E (2025) Peer Review #4 of "FLASC: a flare-sensitive clustering algorithm (v0.1)". PeerJ Computer Science

---

## Round 0.2 · accepted · Accept

The reviewers have indicated that the revised manuscript addressed their concerns. Based on these inputs as well as the scrutiny of your response to the previous reviews, the current manuscript is deemed adequately improved for acceptance.

Reviewer 3 ·

Basic reporting

As my comments have been addressed, I agree to accept the manuscript.

Experimental design

As my comments have been addressed, I agree to accept the manuscript.

Validity of the findings

As my comments have been addressed, I agree to accept the manuscript.

Cite this review as
Anonymous Reviewer (2025) Peer Review #3 of "FLASC: a flare-sensitive clustering algorithm (v0.2)". PeerJ Computer Science

·

Basic reporting

Self-contained with relevant results to hypotheses.

Experimental design

Methods described with sufficient detail & information to replicate

Validity of the findings

Conclusions are well stated, linked to original research question & limited to supporting results

Additional comments

The authors have correctly addressed the observations I made in the first review.

Cite this review as
Rendón lara E (2025) Peer Review #4 of "FLASC: a flare-sensitive clustering algorithm (v0.2)". PeerJ Computer Science